# Health Policies for Rare Disease Patients: A Scoping Review

**DOI:** 10.3390/ijerph192215174

**Published:** 2022-11-17

**Authors:** Luís Carlos Lopes-Júnior, Victor Evangelista Faria Ferraz, Regina Aparecida Garcia Lima, Sara Isabel Pimentel Carvalho Schuab, Raphael Manhães Pessanha, Geisa Santos Luz, Mariana Rabello Laignier, Karolini Zuqui Nunes, Andressa Bolsoni Lopes, Jonathan Grassi, Juliana Almeida Moreira, Fabrine Aguilar Jardim, Franciéle Marabotti Costa Leite, Paula de Souza Silva Freitas, Silvia Regina Bertolini

**Affiliations:** 1Health Sciences Center, Graduate Program in Public Health, Federal University of Espírito Santo (UFES), Vitoria 29047-105, ES, Brazil; 2Ribeirão Preto Medical School, University of São Paulo (USP), Ribeirão Preto 14049-900, SP, Brazil; 3Ribeirão Preto College of Nursing (USP), University of São Paulo, Ribeirão Preto 14040-902, SP, Brazil; 4Nursing Department, Federal University of Espírito Santo (UFES), Vitoria 29047-105, ES, Brazil; 5Hospital de Clínicas da Faculdade de Medicina de Marília (HCFAMEMA), Marília 17519-080, SP, Brazil; 6Department of Integrated Health Education, Graduate Program in Nutrition and Health, Federal University of Espírito Santo (UFES), Vitoria 29043-213, ES, Brazil; 7Department of Management and Health Care, Federal University of São Paulo (UNIFESP), São Paulo 04021-001, SP, Brazil

**Keywords:** rare diseases, public health, health policy, health management

## Abstract

Objective: To identify and map the available evidence on the implementation of public health policies directed at individuals with rare diseases, and to compare the implementation of these health policies between Brazil and other countries. Method: A scoping review guided by the PRISMA-ScR and JBI checklists. The search for articles was conducted in eight electronic databases, MEDLINE/Pubmed, Embase, Cochrane Library, Web of Science, Scopus, CINAHL, PsycINFO, and LILACS, using controlled descriptors, synonyms, and keywords combined with Boolean operators. All steps of this review were independently conducted by two researchers. The selected studies were classified by evidence hierarchy, and a generic quantitative tool was used for the assessment of the studies. Results: A total of 473 studies were identified, of which 13 which met all the inclusion criteria were selected and analyzed. Of these studies, 61.5% (n = 8) had final scores equal to or greater than 70%, i.e., they were classified by this tool as being well-reported. The comparative analysis of international rare diseases demonstrates that public authorities’ priorities and recommendations regarding this topic also permeate and apply to the Brazilian context. Conclusions: The evaluation and monitoring of public policies directed at rare disease patients are urgent and necessary to improve and implement such policies with less bureaucracy and more determination for this unique population that requires timely and high-quality care.

## 1. Introduction

Interest in rare diseases has increased in recent years, resulting in their recognition as a global public health problem with high biopsychosocial and economic burden to patients, families, and health systems [1,2]. Thus, public policies have been developed and implemented in several health systems around the world to meet the needs of these patients by allowing improved access to care [3,4,5,6].

According to the World Health Organization (WHO), a rare disease is defined as a condition that affects less than 65 out of every 100,000 people [7]. At least 7400 rare diseases are listed by the National Institutes of Health and the National Organization of Rare Disorders [8,9]. In general, rare diseases can be placed in a maximum prevalence range of 0.5–7 per 10,000 population. This information is essential to define the scope of official policies developed by each nation [10]. Although rare diseases individually affect a small number of people, collectively they affect between 6–8% of the world population, or 420–560 million people [8,9,11].

Service access and consumption by people with rare diseases are below expectations, which is a clear example of the divergence between the constitutional proposition and the reality of specialized genetic and/or reference services for these conditions [12,13].

A previous review aimed to examine and compare published reports on national plans, policies, and legislation related to all rare diseases in different countries from January 2000 to December 2017, by searching Google and Google Scholar, PubMed, and the Orphanet and National Organization for Rare Disorders (NORD) websites to obtain information on 23 countries. In these countries, rare diseases were found to be defined similarly, differently, or had no definition. The authors concluded that multinational programs supported by common or similar laws are more likely to have a greater impact on rare diseases than single-country programs [14].

Nevertheless, providing adequate assistance to rare disease patients requires rethinking policies to combine the two main facets involved: one for care and treatment, and the other for orphan drug provision [15]. A small percentage of rare diseases have available drug treatments capable of interfering with their progression (orphan drugs), but the high cost of these drugs has often required governments to implement specific policies and procedures to ensure their continued supply through judicial reviews [3,11,15]. In practice, this binomial requires the organization of a service network that combines high-tech therapies with low-complexity procedures, in addition, infrastructure consistent with the patients’ health needs, access to drugs, and treatment monitoring to meet the demands of this population are all required [10,11,15].

As 95% of rare diseases have no treatment and depend on a palliative care network to guarantee or improve the quality of life of patients, this challenge has become quite complex [3,4,16]. Thus, the need to reflect on public policies for the population affected by rare diseases worldwide is evident, and the need to ensure healthcare access and quality is urgent [17]. We hypothesize that the implementation and regulation of public health policies for people with rare diseases vary substantially across countries depending on their income level.

In this context, the objective of this study was to: (i) identify and map the available evidence on the implementation of public health policies directed at rare disease patients, and (ii) to compare the implementation of health policies for people with rare diseases between Brazil and other countries.

## 2. Materials and Methods

### 2.1. Study Design

A scoping review [18,19] following the Preferred Reporting Items for Systematic Review and Meta-Analyses extension for Scoping Reviews (PRISMA-ScR) [20] [Table A1], and in line with the JBI Manual for Evidence Synthesis [19] was performed in accordance with the following steps: (1) definition and alignment of the objectives and research question; (2) definition of the inclusion criteria according to the objective(s) and the guiding question; (3) description of the planned approach for evidence search, selection, data extraction, and evidence presentation; (4) search for evidence; (5) evidence selection; (6) evidence extraction; (7) evidence analysis; (8) result presentation; and (9) evidence synthesis in consideration of the objective, conclusions, and implications of the review. This review is registered with an Open Science Framework under the registration number: osf.io/7kyxu.

### 2.2. Research Question

The Population, Concept, Context (PCC) mnemonic strategy [19] was used to construct the research guiding question, where P indicates rare disease patients, C indicates Comprehensive Care, and C indicates Public Health Policies. Thus, the guiding question was: What evidence is available on the implementation of public health policies aimed at people with rare diseases?

### 2.3. Search Strategy

In the present scoping review, the search for studies was conducted in eight different electronic databases: (I) Medical Literature Analysis and Retrieval System Online (MEDLINE) via PubMed; (II) Excerpta Medica (EMBASE); (III) Cochrane Library; (VI) SCOPUS; (V) Web of Science and (VI) PsycINFO; (VII) Cumulative Index to Nursing and Allied Health Literature (CINAHL); and (VIII) Latin American and Caribbean Health Sciences Literature (LILACS). In addition to the above-mentioned electronic databases, secondary searches were performed in sources such as Clinical Trials Registry websites (ClinicalTrials.gov (National Institutes of Health, NIH, Bethesda, Maryland, USA), WHO International Clinical Trials Registry Platform), Google Scholar, and The British Library. The references of the primary studies were further manually analyzed for additional relevant studies. To locate the articles, descriptors related to the research topic were selected based on the PCC strategy. Then, Medical Subject Headings (MeSH), Health Science Descriptors (DeCS), Emtree, CINAHL headings, and PsycINFO thesaurus were used to verify whether these words were controlled or uncontrolled descriptors were indexed in these bases, using the Boolean operators AND and OR [21,22]. The complete search strategy used in all databases and additional sources is presented in Table A2.

### 2.4. Inclusion and Exclusion Criteria

The inclusion criteria were: (i) original articles indexed in the selected databases on health policy implementation and operationalization for rare disease patients worldwide; (ii) published in the last 21 years (2000–2020); (iii) in Portuguese, English, or Spanish. The exclusion criteria were: (i) review studies, guidelines, editorials, dissertation, thesis, and opinion studies or experience reports; or studies not including the outcome policy implementation and/or operationalization. 

At this stage of the review, the EndNote™ reference manager was used to store and organize papers and exclude duplicates, ensuring a systematic, comprehensive, and manageable search. The studies retrieved from the respective databases and exported to EndNote™ were later imported into the Rayyan™ application, a tool to assist article selection, especially in the phase of study eligibility and inclusion, developed by the Qatar Computing Research Institute. The sample was blindly selected and updated by two independent reviewers. After this selection, a third reviewer analyzed and decided in conjunction with the other two reviewers on the inclusion or exclusion of each article, especially the conflicting ones, using the Rayyan™ App [23].

### 2.5. Data Extraction

Pre-established tools were used for data extraction [21,22,23,24,25], which included four domains: (a) study identification with article title, country of authors, year of publication, host institution, conflict of interests, and funding; (b) methodological characteristics, including study design, objective or research question or hypotheses, sample characteristics (sample size and age, baseline characteristics of experimental and control groups), recruitment method, losses, follow-up duration, and statistical analysis; (c) main findings and implications for clinical practice; and (d) conclusions [21,22,23,24,25]. The selected studies were then classified by evidence hierarchy [26]. This classification was chosen as it is a widely used and effective strategy to classify scientific evidence for systematic literature reviews in the health field. This classification system is divided into seven hierarchical levels, as described in Table 1. Here we have considered levels I to III as strong, IV to VI as moderate, and VII as weak.

This step was followed by the methodological evaluation of the studies. At this stage, the articles were fully read, and their contents were analyzed using the generic quantitative assessment tool developed by Law et al. [27], which includes 12 criteria that represent key elements for evaluating the methodological quality of studies. Each statement of the tool was scored as 1 or 0 and the overall score was calculated by summing the scores and converted into percentages for interpretation. A study with a score of 100% does means that it is a methodologically very well-reported [27]. It should be highlighted that the scores for each study were independently assessed by two reviewers, both nurses, who hold Ph.D.s and had expertise in the subject of rare diseases as well as in review methods, in an independent and blind manner. The disagreements were resolved by a third reviewer, also a nurse, Ph.D., and a full professor with expertise in rare diseases and review studies.

### 2.6. Data Synthesis

The data extracted from the articles that are part of the final sample of this scope review were descriptively analyzed by outcomes to portray the research objectives.

## 3. Results

The search stage identified 473 studies in the eight selected databases. Of these, 13 duplicates were excluded by EndNote™. The selection phase proceeded, with 460 articles meeting the inclusion and exclusion criteria managed by the Rayyan App™. At this stage, 390 articles were excluded by title and abstract reading. Subsequently, the full texts of the remaining 70 articles were read in full. Finally, 13 studies met all the eligibility criteria, and were therefore included and analyzed. Figure 1 shows the study selection flowchart of this scoping review.

Table 2 shows a characterization of the included studies by descending year of publication.

Regarding the methodological quality of the 13 studies, based on the generic quantitative assessment tool [27,28], 61.5% of the studies (n = 8) had final scores equal to or greater than 70%, i.e., from the methodological point of view, they were classified as well reported by this tool. However, four studies (38.5%, n = 5) presented final scores between 50–60%, and were thus designated as having a moderate reporting level (Table 3).

## 4. Discussion

The recognition of access to health as a fundamental social right constitutes an evolution of public policies, and will further contribute to the identification of new roles for the State in the field of healthcare, making it part of the list of government obligations. This process will result in a global increase in public and private spending on health. These costs are not only related to the incorporation of new technologies, but also apply to the ethical commitment to extend coverage to segments still unassisted and to reduce access inequalities [41].

This situation is aggravated by the issue of low-prevalence diseases, termed rare diseases, which, in the context of the pharmaceutical industry, make the process of new orphan drug research and production difficult and expensive. This high expense and low benefit can make it difficult for the government to choose between the best use of public resources and the best health outcomes for patients [10]. In addition to these factors, many rare diseases still lack effective and safe treatments, while even for diseases with treatments, there are several barriers to accessing orphan drugs [41]. This makes the search for health expenditure rationalization an essential issue for democratic governments and public policymakers, with the decision involving a trade-off between ensuring the universality, integrality, and equity of the health system and allocating scarce resources between different health demands with efficiency, efficacy, and effectiveness in reducing public health costs [41].

In the present scoping review, seven of the thirteen analyzed studies were Brazilian, which draws attention to the current discussions on health policies for rare disease patients in Brazil, which have mobilized the scientific community studying the topic [42,43]. Brazil lives under the aegis of the 1988 Constitution, which represents a milestone in recognizing social rights. More than 30 years after its promulgation, the expansion of access to these rights is undeniable [12]; however, the Brazilian public health system still has several challenges to overcome. First, it is difficult to implement general, feasible, and equitable norms in such a huge and unequitable country. Second, this discussion deal with topics both technical and procedural, which involves excessive details, bureaucracy, and complexity [12].

Despite the attempts of the Unified Health System (SUS) to minimize inequality and improve healthcare access and integrality, the regulation and operationalization of public policies have remained an imposing obstacle to the health of the population since its creation [12,15].

The limited access to diagnostic and therapeutic resources has led many patients to use judicial reviews based on the prerogatives of the constitutionally guaranteed rule of law [16,42]. This situation reflects, on the one hand, the paternalistic view that society has of the State and, on the other, the lack of clear rules to regulate the care for rare diseases in the country [17].

The formulation of specific public policies for rare diseases in many countries around the world is a modern movement triggered by the effective social action of patients with these disorders and their caregivers. Until the early 1980s, rare disease patients were largely ignored by both government authorities and the pharmaceutical industry. The work of patient organizations and social movements around the world has given voice to the needs of these people and contributed to rare diseases being considered a global public health problem [10]. In Brazil, this movement culminated in the publication of the Brazilian Policy of Comprehensive Care for People with Rare Diseases (BPCCPRD) within the SUS, in a very pluralistic way and with social participation [41,43,44].

The BPCCPRD was established by the Ministry of Health to reduce morbidity and mortality, and improve the quality of life of people with rare diseases. Several laboratory tests, most using molecular genetic technologies, have been incorporated by the Brazilian public health system, and 18 specialized centers have so far been established at university hospitals (UH) in the capitals of the Southern, Southeastern, and Northeastern regions. However, whether the available human and technological resources in these services are appropriate and sufficient to achieve the goals of care established by the BPCCPRD is as yet unknown. The Brazilian Rare Disease Network (BRDN) is currently under development, comprising 40 institutions, including 18 UHs, 17 rare disease reference services, and five newborn screening reference services [43].

Statistics from the WHO have indicated that between 6% and 8% of the world population have a rare disease, which is equivalent to approximately 420–560 million people [7]. According to epidemiological profile data from the Pharmaceutical Research Industries Association (Interfarma) through the analyses of several concepts adopted in the world, rare diseases can be placed in a maximum prevalence range of 0.5–7 per 10,000 population [10].

To the extent that emerging countries such as Brazil can reduce some causes of mortality such as malnutrition and other infectious and parasitic diseases, rare diseases are gaining prominence in the public health scenario [45,46]. However, 95% of rare diseases have no treatment and require specialized rehabilitation services to improve patient quality of life. Only 2% of rare diseases benefit from orphan drugs capable of interfering with disease progression; a further 3% rely on established treatments for common or prevalent diseases, which can help alleviate symptoms [10]. Due to the rarity of these diseases and the restricted consumer market, investments in research and development to produce drugs for their treatment are difficult, expensive, and risky, making this issue not only a public health problem, but also an economic and social problem [47].

Together, these aspects have generated the need for the implementation of legislation defining some points related to rare diseases and incentives for the development of orphan drugs in Brazil [48]. The origin of the discussion on such legislation came with the creation, in 2000, of a working group that drafted a proposal that culminated in the promulgation of the National Policy for Comprehensive Care of Clinical Genetics in the SUS [12]. This policy, despite being published in 2009 [12], has not yet been regulated and/or implemented. In the following decade, demands from the Brazilian Society of Medical Genetics, especially represented by non-governmental organizations and research institutions, which had been discussing the subject for some years, along with sectors of the Ministry of Health, lead to the structuring and publication of the Ordinance that created the BPCCPRD in 2014, incorporating tests and accrediting institutions to care for patients with these conditions [12,43]. Thus, the publication of the BPCCPRD was undoubtedly a major step for SUS towards equal access, welcoming people with rare diseases, reducing morbidity and mortality and secondary conditions, and improving the quality of life of these patients by facilitating prevention, early diagnosis/detection, and multidisciplinary care organized transversely with the existing networks in the system. In addition, the BPCCPRD represented a measure to mitigate the high costs resulting from judicial reviews, in which the State loses its bargaining power due to the urgency of the purchases determined by the lawsuit [41,43].

One of the BPCCPRD’s principles is the availability of drug treatment and nutritional metabolic formulas, whose incorporation must be based on recommendations evaluated by the CONITEC and the Clinical Protocols and Therapeutic Guidelines Committee (PCDT) [49]. CONITEC was created by Law No. 12,401 in 2011, and provides for therapeutic care and the incorporation of health technology within the SUS. It is a permanent collegiate body, which makes up part of the regimental structure of the Ministry of Health (MOH) whose purpose is to advise the MOH in the attributions related to the incorporation, exclusion, or amendment by the SUS of health technologies, as well as in PCDT constitution or amendments [50].

However, in Brazil, the innovations brought about by Law No. 12,401/2011 [51] have not yet been fully incorporated into the SUS, generating expectations regarding its contribution to improving the SUS and reducing litigation [37,52]. Brazil still does not have a sustainable policy on access to rare diseases. Although the government has expanded the budget for the purchase of drugs, particularly for highly complex diseases, in addition to programs to expand access to drugs for chronic diseases, much that has been conducted remains insufficient to meet the new demands. The population is still generally dissatisfied with the underfunding of the public system, which suppresses demand through artifices such as registration delays, or even by denying the incorporation of new technologies [53].

There is also an increased number of health-related judicial reviews in the country, leading to a considerable financial and social impact [49]. However, these judicial reviews, which shelter the truly needy, also create opportunities for access system disorganization, privileging those who know how to demand and, almost always, being devoid of careful and accurate proof of the safety and effectiveness of treatments and their suitability to the one who files the review [5].

The comparison of international rare disease and orphan drug regulations shows that the priorities of public authorities and the recommendations on this issue also permeate and apply to the Brazilian context. These priorities include: (a) the consideration of rare diseases as a public health priority and focus on assertive decision-making considering the monitoring of current legislations to confirm the definition and classification of rare diseases; (b) the development of indicators to more accurately feed epidemiological databases; (c) the creation of more incentives for research and investments to encourage the pharmaceutical industry to develop new orphan drugs; (d) the reorganization of national health systems to guarantee access to orphan drugs; (e) the increase in the dissemination of knowledge and information on rare diseases for patients, caregivers, families, and health professionals; (f) the encouragement of the civil movement and popular participation in this process; and (g) increase international cooperation in clinical research on rare diseases and the development of new orphan drugs [31,32,34,35,38,39,40,43,54].

This review has some limitations. First, the time limit established (2000–2020) and the language restrictions (English, Portuguese, and Spanish) may have influenced the final sample of articles. Second, the gray literature and preprints were not considered. Third, most of the evidence gathered here fell into classification VI and was considered moderate. Further well-designed studies with a higher level of evidence (I, II, and III) are recommended to be conducted on the subject.

## 5. Conclusions

It can be concluded that Brazil still does not have a sustainable policy on access to rare diseases. The comparative analysis of international rare disease and orphan drug regulations demonstrates that public authorities’ priorities and recommendations regarding this topic also permeate and apply to the Brazilian context. Furthermore, it ratifies the need to constantly evaluate and monitor public policies directed at rare disease patients so that they can be improved, and their successes can be strengthened and implemented with less bureaucracy and more determination for this unique population that requires timely and high-quality care.

## Figures and Tables

**Figure 1 ijerph-19-15174-f001:**
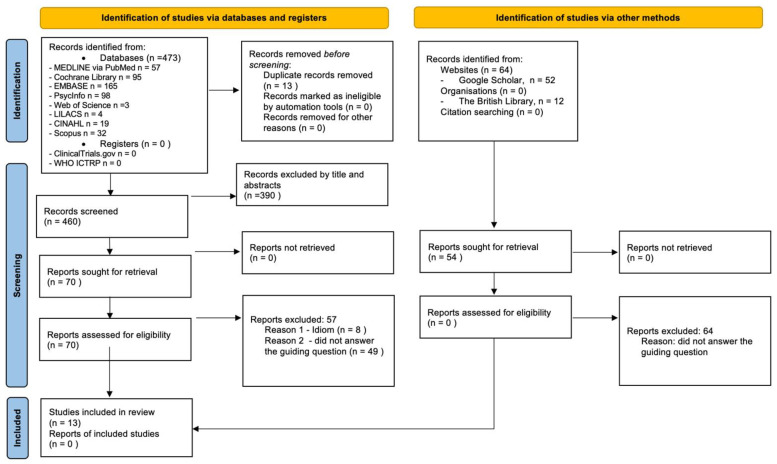
Flowchart of the studies included in the scope review.

**Table 1 ijerph-19-15174-t001:** Evidence hierarchy levels.

Evidence Level	Study Design
I	Evidence from systematic reviews or meta-analyses of randomized controlled trials (RCTs)
II	Evidence derived from a well-designed randomized controlled trial (RCT)
III	Evidence obtained from a well-designed controlled clinical trial without randomization (quasi-experiments)
IV	Evidence from a cohort study, a case-control study, a well-designed cross-sectional study
V	Study from a systematic review of qualitative studies (metasynthesis) and descriptive studies
VI	Evidence derived from a single descriptive or qualitative study
VII	Evidence from the opinion of authorities and/or expert committee reports

**Table 2 ijerph-19-15174-t002:** Characteristics of the 13 selected studies.

Citation	Country	Objectives	Main Results	Design/ Evidence Level
Franciscatto LHG, et al. [28] /2020	Brazil	Identify the health service experiences of families with children with genetic diseases.	The diagnosis of genetic disease causes substantial family changes due to the search for a treatment to meet the child’s needs. Families experienced difficulties such as unpreparedness of health professionals, lack of service organization, litigation of resources, and the need for a structured Health Care System.	Qualitative /VI
Iriart JAB, et al. [29] /2019	Brazil	Analyze the therapeutic itineraries of patients with rare genetic diseases in the cities of Rio de Janeiro, Salvador, and Porto Alegre, focusing on the material, emotional, and structural challenges faced during the search for diagnosis and treatment.	The experience of having a rare genetic disease, besides being a challenge due to its debilitating and disabling characteristics, is aggravated by practical-relational and bureaucratic-institutional issues which are not solved at a specialized service. Long therapeutic itineraries until diagnosis, lack of knowledge on rare diseases by non-geneticist physicians, difficult transport and access to specialists, diagnostic and complementary tests, and high-cost drugs and food supplies were common factors between the three cities.	Qualitative /VI
Lima MADFDD, Gilbert ACB, Horovitz DDG. [30] /2018	Brazil	Investigate how rare disease patient organizations in Brazil manage access to treatment using social media.	The role of patients’ associations is multi-faceted, ranging from advising patients and families on issues related to treatment and quality of life to active participation in the development and implementation of public policies.	Descriptive /VI
DHARSSI S, et al. [31] /2017	United States	Understand opportunities for further policy development by examining how these policies and programs can align with community needs.	Policy status and implementation were assessed for each country in the context of the five dimensions of main patient needs (care management, diagnostic resources, treatment access, patient awareness and support, and innovative research). The continuous role of the community in directing and implementing legislation and programs to improve rare disease care is crucial. The implementation of rare disease care plans is very uneven across countries.	Descriptive /VI
Gong S, et al. [32] /2016	China	Assess the availability and affordability of orphan drugs in China.	Orphan drugs approved in the US, EU, and Japan had 37.8%, 24.6%, and 52.4% of the market availability in China, respectively. The mean availability of 31 orphan drugs surveyed in the 24 tertiary public hospitals in China was 20.8%. Within a periodic course of treatment, the mean treatment cost of 23 orphan drugs was approximately USD 4,843.5. Twenty-two orphan drugs for 14 rare diseases were unaffordable to most residents in China.	Descriptive /VI
Melo DG, et al. [33] /2015	Brazil	Analyze the genetic competencies of primary health care professionals in Brazil.	Regarding knowledge, about 80% of the participants recognized basic genetic terminology, but had difficulty identifying inheritance patterns. Regarding clinical skills, physicians were able to recognize facial dysmorphisms and identify in which situations to refer patients to specialists.	Descriptive /VI
Maresova P, Mohelská H, KUCA K. [34] /2015	European Union and the Czech Republic	Analyze the relationship between the economic level of the European Union and the Czech Republic through their development index, gross domestic product spending on health, and expenditure on orphan drugs for patients with rare diseases.	The comparison of the evolution of basic macroeconomic variables showed that the EU and the Czech Republic had almost identical expenditures on orphan drugs. Thus, the hypothesis of higher expenditure on orphan drugs in countries with a higher economic status was not confirmed. In recent years, the Czech Republic has presented concerns about the increased costs of these drugs. Forecasts for EU countries and the Czech Republic show a stagnation of orphan drug costs. Conversely, increased costs are expected in countries where these expenditures are lower than the Euro zone average, e.g., in Sweden, of 2.7%, and in France, of 3.2%.	Descriptive /VI
Garrino L, et al. [35] /2015	Italy	Explore the impact of rare diseases on patients’ lives and the experience of healthcare professionals caring for these patients.	Five topics were identified in the patients’ speeches––“coping with the development of the disease”, “living with the disease”, “daily life”, “relationship with others”, and “relationship with health professionals”.	Qualitative /VI
Luz GS, Silva MRS, DeMontigny [36] /2015	Brazil	Characterize the diagnostic and therapeutic itinerary of families of rare disease patients within the Brazilian public service system.	Three thematic nuclei were identified: “Itinerary of families in the search for the diagnosis of the disease”, “Itinerary of families in the post-diagnosis of the disease”, and “Itinerary of therapeutic maintenance”.	Qualitative /VI
Aith F, et al. [37] /2014	Brazil	Analyze how the Brazilian Committee for Technology Incorporation (CONITEC) promotes the incorporation of new technologies into the SUS and the drug policy for patients with rare diseases considering the current legal configuration.	In Brazil, the principles of universality and integrality are difficult to implement when confronted with the formal processes of new technology incorporated into the SUS. In this sense, judicial review is and will always be a crucial tool to access services and products not incorporated into the SUS for patients with dissimilar needs.	Descriptive /VI
Taruscio D, et al. [38] /2013	European Union	To describe the development of a multicenter project (EUROPLAN) to support the development of national and strategic plans for rare diseases in Europe.	EUROPLAN aimed to promote the implementation of national plans and strategies to address rare diseases and to share relevant experiences between member countries, linking national efforts through a common European strategy. The project was launched in 2008 and involved two implementation phases: phase 1 (2008–2011), to build the consensus definition of operational tools (recommendations and indicators); and phase 2 (2012–2015), which mainly focused on capacity building with the active participation of several level stakeholders. EUROPLAN aims to facilitate and accelerate the implementation of national plans in EU countries.	Descriptive /VI
Diniz D, Medeiros M, Schwartz IVD. [39] /2013	Brazil	Analyze the financial dimension of the judicial review of three high-cost drugs available in the pharmaceutical market for the treatment of mucopolysaccharidoses (MPS) types I, II, and VI in Brazil.	A judicial review of the drugs laronidase, idursulfase, and galsulfase was requested by 195 people in 196 lawsuits, with a total cost of BRL 219,664,476, distributed as follows: BRL 9,262,981 for laronidase and 24 patients with MPS I; BRL 86,985,457 for idursulfase and 68 patients with MPS II; and BRL 123,416,039 for galsulfase and 103 patients with MPS VI (104 lawsuits). Inequality was higher for idursulfase and laronidase, but was also high for galsulfase.	Descriptive /VI
Seoane-Vazquez E et al. [40] /2008	United States	Analyze the characteristics of orphan drug designations, approvals, and sponsors, and to evaluate the patent and market exclusivity of new orphan molecular entities approved in the United States between 1983 and 2007.	The Food and Drug Administration (FDA) listed 1,793 orphan designations and 322 approvals between 1983 and 2007. Cancer was the leading disease group targeted for orphan approvals. Eighty-three companies held 67.7% of total orphan new molecular entity (NME) approvals. The average time from orphan drug designation to FDA approval was 4.0 ± 3.3 years. The average patent term and maximum effective market exclusivity were 11.7 ± 5.0 years for orphan NMEs. Orphan drug market exclusivity increased the average effective patent and market exclusivity life by 0.8 years.	Cohort /IV

**Table 3 ijerph-19-15174-t003:** Methodological evaluation of the included studies.

Reference	Criteria *													
	1	2	3	4	5	6	7	8	9	10	11	12	Scoring	%
[28]	Y	Y	Y	Y	N	NA	NA	N	Y	Y	NR	Y	7/10	70
[29]	Y	Y	Y	Y	N	NA	NA	N	Y	Y	NR	Y	7/10	70
[30]	Y	Y	Y	N	N	NA	NA	NA	Y	Y	NR	Y	6/10	60
[31]	Y	Y	Y	Y	N	NA	NA	N	Y	Y	NR	Y	7/10	70
[32]	Y	Y	Y	Y	N	NA	NA	N	Y	Y	NR	Y	7/10	70
[33]	Y	Y	Y	Y	N	NA	NA	Y	Y	Y	NR	Y	8/10	80
[34]	Y	Y	NR	N	N	NA	NA	N	Y	Y	NR	Y	5/10	50
[35]	Y	Y	Y	Y	N	NA	NA	N	Y	Y	NR	Y	7/10	70
[36]	Y	Y	Y	Y	N	NA	NA	N	Y	Y	NR	Y	7/10	70
[37]	Y	Y	Y	N	N	NA	NA	N	Y	Y	NR	Y	6/10	60
[38]	Y	Y	N	N	N	NA	NA	N	Y	Y	NR	Y	5/10	50
[39]	Y	Y	Y	NR	N	NA	NA	N	Y	Y	NR	Y	6/10	60
[40]	Y	Y	Y	N	N	NA	NA	Y	Y	Y	NR	Y	7/10	70

* Criteria: 1 = Study objective; 2 = Relevant background; 3 = Sample Description; 4 = Sample size justification; 5 = Reliability and Validity of outcome measures; 6 = Description of the intervention; 7 = Contamination and co-intervention; 8 = Statistical significance; 9 = Appropriate analyses; 10 = Clinical-Epidemiological Significance; 11 = Reported drop-outs; 12 = Appropriate conclusions. *N = No; NA= Not Applicable; NR= Not Reported; Y= Yes. Classification of the study: ≥70% = Good quality; ≥50% and <70% = Moderate quality; <50% = Poor quality.

## Data Availability

Not applicable.

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
