# Peer review of "Health Policies for Rare Disease Patients: A Scoping Review"

_ijerph, 2022, doi:10.3390/ijerph192215174_

Round 1

Reviewer 1 Report

This paper is a well-constructed and clear research. It is an interesting contribution that is useful for the scientific community. The introduction is up to date and, although it lacks hypotheses (something I think it would be good to have), it allows understanding and contextualizing the research. 

The researchers analyze a large number of databases to give greater consistency to the review. The research design is correct, the inclusion and exclusion criteria are clearly shown, as well as the data extraction. However, at this point the authors show a hierarchy of evidence, based on a methodological evaluation. External evaluators have been involved in this evaluation. It would be useful to expand the information to better understand how this evaluation was carried out and what expert knowledge these evaluators have. On the other hand, the reason for taking this step in the methodology should be better justified. It is not well understood why this evaluation has been carried out and how it has been carried out by the evaluators.

In the results section we see that there are changes in the typeface used, and mismatches in the tables (Table 3 and 4). When the results of the methodological evaluation are indicated, it would be useful to describe the table a little more. In this regard, it would also be good to explain the meaning of the final score. On the other hand, I believe that it would help other researchers if a brief summary of the articles were included. Table 2 expresses the main results, but it would be good to include the main ideas contributed by these 13 articles. I am referring to something general. That is, a paragraph explaining the general aspects of these 13 papers. In this sense, it would possibly be interesting to show differences between the papers with a higher methodological evaluation and those with a lower evaluation.

I consider that the discussion is well presented and developed.

The conclusions should be expanded further, showing a clear relationship with the objective of the study and explaining where the results obtained lead.

In short, this is a good quality work that needs some adjustments to be clearer and more understandable. In this sense, I would like to congratulate the researchers. Without any doubt, it will be a work to be taken into account for all of us who are doing research on rare diseases. 

Author Response

Reviewer 1

Comments and Suggestions for Authors

- This paper is a well-constructed and clear research. It is an interesting contribution that is useful for the scientific community. The introduction is up to date and, although it lacks hypotheses (something I think it would be good to have), it allows understanding and contextualizing the research.

Response: Thank you so much for your valuable comments as well as positive feedback. As per suggested we have added a hypothesis: “We hypothesize that the implementation and regulation of public health policies for people with rare diseases varies substantially across countries depending on their income level”.

- The researchers analyze a large number of databases to give greater consistency to the review. The research design is correct, the inclusion and exclusion criteria are clearly shown, as well as the data extraction.

Response: Thank you so much!

- However, at this point the authors show a hierarchy of evidence, based on a methodological evaluation. External evaluators have been involved in this evaluation. It would be useful to expand the information to better understand how this evaluation was carried out and what expert knowledge these evaluators have. On the other hand, the reason for taking this step in the methodology should be better justified. It is not well understood why this evaluation has been carried out and how it has been carried out by the evaluators.

Response: Thank you so much for this important comment! In fact, in scope reviews, methodological evaluation is optional in accordance with the JBI Manual for Evidence Synthesis 2021. In addition, the classification of the evidence obtained is usually using some hierarchy of evidence scale based solely on the study design, followed or not by the evaluation of the report of the included studies through a generic tool. It is worth mentioning that unlike a systematic review in which it is necessary to assess the risk of bias between studies, in a scoping review this approach is not always recommended, but optional.

We added more detail to the paragraph to make it clearer as per recommended.

“This step was followed by the methodological evaluation of the studies. At this stage, the articles were fully read, and their contents were analyzed using the generic quantitative assessment tool developed by Law et al. [27], which includes 12 criteria that represent key elements for evaluating the methodological quality of studies. Each statement of the tool was scored as 1 or 0 and the overall score was calculated by summing the scores and converted into percentages for interpretation. A study with a score of 100% does means that it is a methodologically very well-reported [27]. It should be highlighted that the scores for each study were independently assessed by two reviewers, both nurses, hold Ph.D. and with expertise in the subject of rare diseases as well as in reviews methods, in an independently and blindly manner. The disagreements were resolved by a third reviewer, also a nurse, Ph.D., and full professor with expertise in rare diseases and review studies”.

- In the results section we see that there are changes in the typeface used, and mismatches in the tables (Table 3 and 4). When the results of the methodological evaluation are indicated, it would be useful to describe the table a little more. In this regard, it would also be good to explain the meaning of the final score. On the other hand, I believe that it would help other researchers if a brief summary of the articles were included. Table 2 expresses the main results, but it would be good to include the main ideas contributed by these 13 articles. I am referring to something general. That is, a paragraph explaining the general aspects of these 13 papers. In this sense, it would possibly be interesting to show differences between the papers with a higher methodological evaluation and those with a lower evaluation.

Response: We added more details of this methodological assessment in the methods section and added information in the legend of Table 3. Thanks!

- I consider that the discussion is well presented and developed.

Response: Thank you so much!

- The conclusions should be expanded further, showing a clear relationship with the objective of the study and explaining where the results obtained lead.

Response: Thank you so much! So, we have added a specific objective for this review in order to make more consistent with our results. “In this context, the objectives of this study were: i) to identify and map the available evidence on the implementation of public health policies directed at rare disease patients; and ii) to compare the implementation of health policies for people with rare diseases between Brazil and other countries”.

- In short, this is a good quality work that needs some adjustments to be clearer and more understandable. In this sense, I would like to congratulate the researchers. Without any doubt, it will be a work to be taken into account for all of us who are doing research on rare diseases.

Response: Thank you so much for all comments! Certainly, helped us to improve our manuscript substantially! We really appreciated it!

Reviewer 2 Report

The objective of the paper is to identify and map the available evidence on implementation of public health policies directed at rare disease patients. Its main contribution is the presentation of regulative measures for rare diseases and orphan medicine for a range of countries. Its strengths are the discussion of Brazilian policies towards the same. The method is a scoping study, and results are found for several countries including the EU and US. 

The paper comments on a relevant topic of public policy towards the health domain. Nevertheless, this reader questions the connection between the objective of the article, the presentation of results, the table on country regulations on page 9, the discussion, and the conclusion. That is, the read thread is not easy to follow, and the conclusion do not comment upon findings from the result paragraph nor on the contextualisation of the paper` research question. This may pertain to the fact that the research question is very wide pointing to “identify and map the available evidence on implementation of public health policies directed at rare disease patients”. Scoping studies can operate a more specific research question and this reader suggests that the paper should treat the comparison of regulations between Brazil and other countries.

The connection between the result presentation and table 4 on regulations are specifically unclear. Regulations are the result of public policies, while the results from the scoping study points to the drug market, available knowledge on genetic diseases, expenditures and costs, and lawsuits, to mention some. Further, table 4 includes knowledge on the countries Australia, Japan, South Korea and Taiwan, but articles covering these countries are not included in the scoping study nor in the reference list. Therefore, the table on page 9 is not transparent. The table is interesting, but from what evidence it emanates and why it is presented in addition to the results from the scoping study is unclear. 

To make a comparison between Brazil and other countries on the article` topic can provide interesting and necessary knowledge on Brazilian conditions, as well as of public policies on the topic in the countries Brazil is compared to. As it is, this is a secluded aim of the paper not to be revealed until the discussion. This should be spelled out in the introduction. 

Methodologically, the article seems to be sound. This reader thus, have some questions and the first is: Why is a scoping study design adopted? This seems not to be justified and a narrative literature review could have been at least as interesting to the reader as it makes room for justifying the choices made by authors. Secondly, most articles included in this review are given the label descriptive and ranked sixth in table 1. What does this ranking of evidence levels implicate for this specific study? As for articles included in the study, reliability and validity (item 5 in table 3) is not really discussed. Thirdly, table 3 is a table this reader finds not well justified and transparent. Readers gets to know the authors judgment on the qualities of included articles, but we do not get to know anything about the considerations made by researchers. Adding to the confusion references for these methodological questions are made to other articles from the first author (21-25 and 27), not to literature on methodological questions. Are then the pre-established tools (line 138) only established by the first author? Or are readers expected to also read these articles to understand what this is about? This reader answered no on the question of inappropriate self-citations, nevertheless, as is shown, whether such are made can be questioned.

Author Response

Reviewer 2

Comments and Suggestions for Authors

- The objective of the paper is to identify and map the available evidence on implementation of public health policies directed at rare disease patients. Its main contribution is the presentation of regulative measures for rare diseases and orphan medicine for a range of countries. Its strengths are the discussion of Brazilian policies towards the same. The method is a scoping study, and results are found for several countries including the EU and US.

Response: Thank you so much for your comments and positive feedback.

- The paper comments on a relevant topic of public policy towards the health domain. Nevertheless, this reader questions the connection between the objective of the article, the presentation of results, the table on country regulations on page 9, the discussion, and the conclusion. That is, the read thread is not easy to follow, and the conclusion do not comment upon findings from the result paragraph nor on the contextualisation of the paper` research question. This may pertain to the fact that the research question is very wide pointing to “identify and map the available evidence on implementation of public health policies directed at rare disease patients”. Scoping studies can operate a more specific research question and this reader suggests that the paper should treat the comparison of regulations between Brazil and other countries.

Response: Thank you so much for this valuable comment. We totally agree with you. So, we have added a specific objective for this review in order to make more consistent with our results. “In this context, the objectives of this study were: i) to identify and map the available evidence on the implementation of public health policies directed at rare disease patients; and ii) to compare the implementation of health policies for people with rare diseases between Brazil and other countries”.

- The connection between the result presentation and table 4 on regulations are specifically unclear. Regulations are the result of public policies, while the results from the scoping study points to the drug market, available knowledge on genetic diseases, expenditures and costs, and lawsuits, to mention some. Further, table 4 includes knowledge on the countries Australia, Japan, South Korea and Taiwan, but articles covering these countries are not included in the scoping study nor in the reference list. Therefore, the table on page 9 is not transparent. The table is interesting, but from what evidence it emanates and why it is presented in addition to the results from the scoping study is unclear.

Response: Thank you very much for this careful review and for your valuable contribution to improving the presentation of our article. In fact, this table was displaced and we agree with you.  These data are compiled from another  review published in 2012 on the subject of regulation of rare diseases (but which was not included in the sample of our study, as one of the exclusion criteria was "review studies". Hence, We dicided to remove this table, so that our text is more fluid and consistent. Thanks for the suggestion!

- To make a comparison between Brazil and other countries on the article` topic can provide interesting and necessary knowledge on Brazilian conditions, as well as of public policies on the topic in the countries Brazil is compared to. As it is, this is a secluded aim of the paper not to be revealed until the discussion. This should be spelled out in the introduction.

Response: Thank you so much for this valuable comment. We totally agree with you. So, we have added a specific objective for this review (in the introduction section as per suggested) in order to make more consistent with our results. “In this context, the objectives of this study were: i) to identify and map the available evidence on the implementation of public health policies directed at rare disease patients; and ii) to compare the implementation of health policies for people with rare diseases between Brazil and other countries”.

- Methodologically, the article seems to be sound. This reader thus, have some questions and the first is: Why is a scoping study design adopted? This seems not to be justified and a narrative literature review could have been at least as interesting to the reader as it makes room for justifying the choices made by authors. Secondly, most articles included in this review are given the label descriptive and ranked sixth in table 1. What does this ranking of evidence levels implicate for this specific study? As for articles included in the study, reliability and validity (item 5 in table 3) is not really discussed. Thirdly, table 3 is a table this reader finds not well justified and transparent. Readers gets to know the authors judgment on the qualities of included articles, but we do not get to know anything about the considerations made by researchers.

Response: We have opted for a scope review because it is a consolidated method in the literature with a specific guideline to guide (JBI- which we are following), that is, it has a methodological rigor in the conduction, mainly with regard to the search strategy in the several databases (essential databases, specific databases and complementary databases), with clear inclusion and exclusion criteria; quite different from a narrative review of the literature that does not have any method, the authors can even do it from a single database and without criteria in the search strategy and with a lot of subjectivity in the conduction. That's why we chose a scope review in order to qualify the importance of the theme under a consistent method.

Yes, most of the studies included in this scoping review have a descriptive design and according to the hierarchy of evidence we used, they are considered moderate evidence.

We added this information to the method: "Here we have considered levels I to III as strong, IV to VI as moderate, and VII as weak." In addition, we added this in the limitation and recommend that new studies with a higher level of evidence (I, II and III) be conducted on the subject.

In fact, in scope reviews, methodological evaluation is optional in accordance with the JBI Manual for Evidence Synthesis 2021. In addition, the classification of the evidence obtained is usually using some hierarchy of evidence scale based solely on the study design, followed or not by the evaluation of the report of the included studies through a generic tool. It is worth mentioning that unlike a systematic review in which it is necessary to assess the risk of bias between studies, in a scoping review this approach is not always recommended, but optional.

We added more detail to the paragraph to make it clearer as per recommended.

“This step was followed by the methodological evaluation of the studies. At this stage, the articles were fully read, and their contents were analyzed using the generic quantitative assessment tool developed by Law et al. [27], which includes 12 criteria that represent key elements for evaluating the methodological quality of studies. Each statement of the tool was scored as 1 or 0 and the overall score was calculated by summing the scores and converted into percentages for interpretation. A study with a score of 100% does means that it is a methodologically very well-reported [27]. It should be highlighted that the scores for each study were independently assessed by two reviewers, both nurses, hold Ph.D. and with expertise in the subject of rare diseases as well as in reviews methods, in an independently and blindly manner. The disagreements were resolved by a third reviewer, also a nurse, Ph.D., and full professor with expertise in rare diseases and review studies”.

- Adding to the confusion references for these methodological questions are made to other articles from the first author (21-25 and 27), not to literature on methodological questions. Are then the pre-established tools (line 138) only established by the first author? Or are readers expected to also read these articles to understand what this is about? This reader answered no on the question of inappropriate self-citations, nevertheless, as is shown, whether such are made can be questioned.

Response: Thanks for your comments. Methodological assessment tool is totally different from extraction form. Here we are talking about two different things! One thing is the 12-criteria methodological assessment tool we use.” the generic quantitative assessment tool developed by Law et al. [27] which includes 12 criteria that represent key elements for evaluating the methodological quality of studies”. The other, totally different, is related to the extraction form, which is one of the stages of the review studies and also of the scope review. Overall, such extraction forms are prepared by the researchers themselves or may be based on items from previously published forms (in our case). Thus, we chose to use the Extract Forms previously prepared by the first author who is an expert in review studies with systematized methods (systematic review, scope review, integrative review and meta-analyses), in addition to being a professor of Evidence-Based Practice and  Associate Review Editor of Frontiers in Medicine (in the area of ​​Review Studies), therefore, he has the know-how to do so and the adequacy of citations is justified.

Yours Sincerely,

The authors

Round 2

Reviewer 1 Report

The changes made by the authors have improved the article. I think the final result is good. Congratulations to the authors.

Reviewer 2 Report

I think my comments are answered to and I have no further comments for authors.